# Role of Egg Parasitoids in Controlling the Pine Processionary Moth in the Cedar Forests of Chréa National Park (Algeria)

**Safia Sebti [1], Luís Bonifácio [2,3,*] and Gahdab Chakali [4]**

1.  Faculté des Sciences de la Nature et de la Vie, Université Saad Dahlab, B.P. 270, Route de Soumâa, Blida 09000, Algeria; sebtisaf@yahoo.fr
2.  Instituto Nacional de Investigação Agrária e Veterinária, Quinta do Marquês, Av. da República, 2780-159 Oeiras, Portugal
3.  GREEN-IT Bioresources for Sustainability, ITQB NOVA, Av. da República, 2780-157 Oeiras, Portugal
4.  Ecole Nationale Supérieure Agronomique, El-Harrach, Alger 16200, Algeria; chakali_gahdab@yahoo.fr
*   Correspondence: luis.bonifacio@iniav.pt

**Abstract:** The sustainable protection of cedar stands in Chréa National Park can only be accomplished through the stability of the ecosystem. Outbreaks of *Thaumetopoea pityocampa* are a major threat and are largely attributed to the high population fecundity, changes in the diversity of natural enemies and global interactions within the ecosystem. Egg parasitoids (Hymenoptera) are essential in the biological control of *T. pityocampa*. To assess the impact of the parasitoids on the populations of the pine processionary moth, egg masses from cedar plantations were collected, reared in a laboratory and checked regularly for the emergence of the egg parasitoids: *Trichogramma embryophagum*, *Baryscapus servadeii* and *Ooencyrtus pityocampa*. Observations showed an inter-annual variation in the abundance of the three parasitoids as a result of the variation in the population density of the processionary moth, and on the underlying effect of temperature. Parasitoids had variable parasitism rates, with yearly averages ranging from 3.86% to 51.14%, dependent on the spatiotemporal distribution of the host populations. The aggregate effect of multiple parasitoid species could optimize control of *T. pityocampa* in cedar stands.

**Keywords:** *Thaumetopoea pityocampa*; *Trichogramma embryophagum*; *Baryscapus servadeii*; *Ooencyrtus pityocampa*; fecundity; interaction; Atlas cedar

## 1. Introduction

Cedar forests are part of the Mediterranean endemic forest ecosystems of the North African mountain areas. In Algeria, the majority of cedar stands are located in National Parks and are in decline since the end of the 20th century mainly due to unfavorable climatic changes [1]. In fact, despite the adaptability of this Mediterranean species, global warming and water deficit affect the resistance in trees to attacks by different defoliators and xylophages [2].

The Pine Processionary Moth (PPM) *Thaumetopoea pityocampa* (Denis and Schiffermuller, 1775) is the most important defoliator pest of pine and cedar in North Africa [3–5] and Southern Europe [6], PPM affects the health status of trees, causing serious defoliation episodes resulting in important losses in growth, productivity [7,8] and even survival of stands [9]. Currently, the PPM is omnipresent in the Chréa cedar forest, with frequent outbreaks [10]. The infestation periods of PPM are linked to the absence of climatic constraints, notably lethal temperatures [11] and to resources availability [12]. Ecological equilibrium reestablishment results from strong natural regulation, based on the interactions between a complex of natural enemies. These interactions are strongly modulated by site conditions and the establishment capacities of the involved species. The intracyclic and transcyclical fluctuations, which characterize the biological cycles of insects generally depend on climatic variations, which can generate a spatiotemporal desynchronization between pests and their

natural enemies [13,14], affecting the natural regulatory processes. On the other hand, the phenomena underlying global warming also have impact on PPM populations [15]. PPM shows a positive response to increases in winter temperatures, which promote high larval survival, and expansion to higher altitude and latitude [16,17] allowing for the colonization of new areas [18,19]. Climatic changes also alter interspecific interactions [20–22]. Some studies [23,24] predict disruptions in processionary moth natural enemy relationships via modifications in the synchrony of their life cycle.

The action of parasitoids depends on the density fluctuations of pest populations, and also the effect of variables influencing the spatiotemporal synchronization [23]. The development of control programs must integrate the ecosystem as a whole in order to establish sustainable management of infestations in protected sites.

The main objective of this study is to improve the knowledge on the natural process of PPM population regulation, and to understand the impact of egg parasitoids and their efficacy in reducing the impact of the moth on cedar stands within the National Park.

## 2. Materials and Methods

### 2.1. Study Site

The Chréa National Park (Atlas Blideen cedar forest) is located 50 km south-west of Algiers (36°26′3.5″ N; 2°53′20.6″ E). It covers an area of 26,587 hectares with mountain tops between 1400 and 1600 m. The relief is very rugged, and the slope sometimes exceeds 60%. Due to well-contrasted vertical stratification of vegetation, three ecotypes characterize the Chréa area: Supra-Mediterranean, Meso-Mediterranean and Thermo-Mediterranean. Average monthly temperature in the Chréa area is 5 °C in winter and 22 °C in summer. The region receives an average of 700 mm of precipitation annually. Snow is probable between January and March, from an altitude of 400 m.

### 2.2. Collection of Biological Material

To evaluate the start of the PPM adult flight period in the cedar forests, 6 green funnel traps lured with specific pheromone and were placed in pairs, with 100 m distance between, separated 200 m along the study area, at the beginning of June of every year. As soon as the first adult moths were caught in the traps, a 500 m × 50 m transect was surveyed to collect the egg masses detected at height between 1.5 and 2.5 m (Figure 1). Weekly surveys were carried out until 100 egg masses were collected. In 2010 and 2011, PPM egg masses were more abundant and therefore only three surveys were required, each being carried out for 3–4 h. In the following years, fewer PPM egg masses were found, more time consuming surveys (5–6 h per survey) were required (six in 2012, 2013 and seven in 2014), and it still was not possible to collect 100 egg masses.

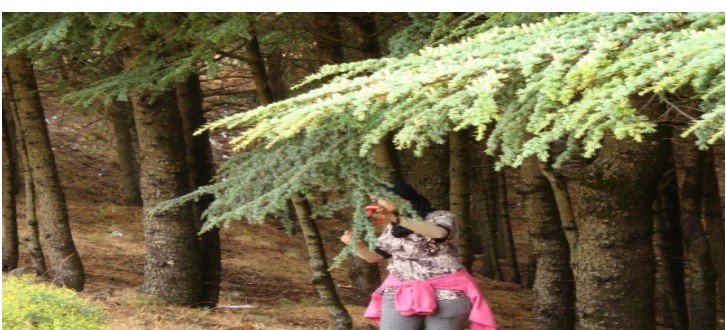 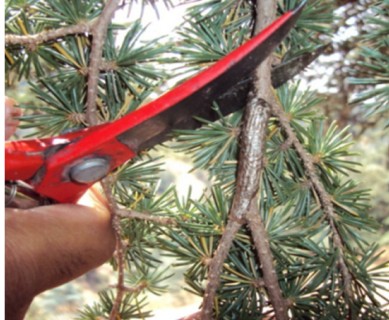

**Figure 1.** Survey and collection of pine processionary moth (*Thaumetopoea pityocampa*) egg mass in twig of cedar in Chréa National Park.

Before storing the egg masses, the length and diameter of each support twig were recorded. Egg masses were kept individually in glass test tubes (1.5 cm diameter and 7.5 cm length) with the opening plugged with cotton. Tubes were numbered and placed under

natural conditions of the forest station located inside the cedar stand in Chréa National Park. Egg masses were checked daily, and emerged parasitoids were obtained and brought to the university laboratory, identified and individually stored. After emergence of all parasitoids and caterpillars, the protective scales (that were detached from the abdomen of the female to cover the eggs) were removed manually using tape and the remaining eggs were examined under a stereomicroscope (magnification × 40). Eggs were categorized as hatched, unhatched or parasitized.

*2.3. Data Analysis*

Air temperature data of the study area used in the analysis were made available by the Algeria National Services, Office National de Météorologie (ONM), Dar El Beïda, Algeria (https://www.meteo.dz/home, accessed on 25 March 2015). The descriptive analysis and the ANOVA, with results interpreted at the error of 5%, were performed on host twig diameters where the eggs were placed, the moth fitness (size of egg masses, fecundity and success of embryonic development) and parasitization rates of the three parasitoids species, along the years of the study. ANOVA with post-hoc LSD for homogeneous groups, was performed using STATISTICA software [25].

**3. Results**

The first PPM adults were caught in the traps in the second half of the month of June, in all years of the study (22 June 2010; 15 June 2011; 21 June 2012; 14 June 2013; and 19 June 2014).

Transect surveys carried out on the cedar stand resulted in the collection of 468 egg masses, during the five years of the study. In 2010 and 2011, three transects were enough to obtain 100 egg masses, with one week delay in 2011 (4–15 July) in relation to 2010 (30 June–8 July). In the following years, fewer egg masses were found and surveys were repeated along all PPM flight periods, without achieving the targeted 100 egg masses (Figure 2).

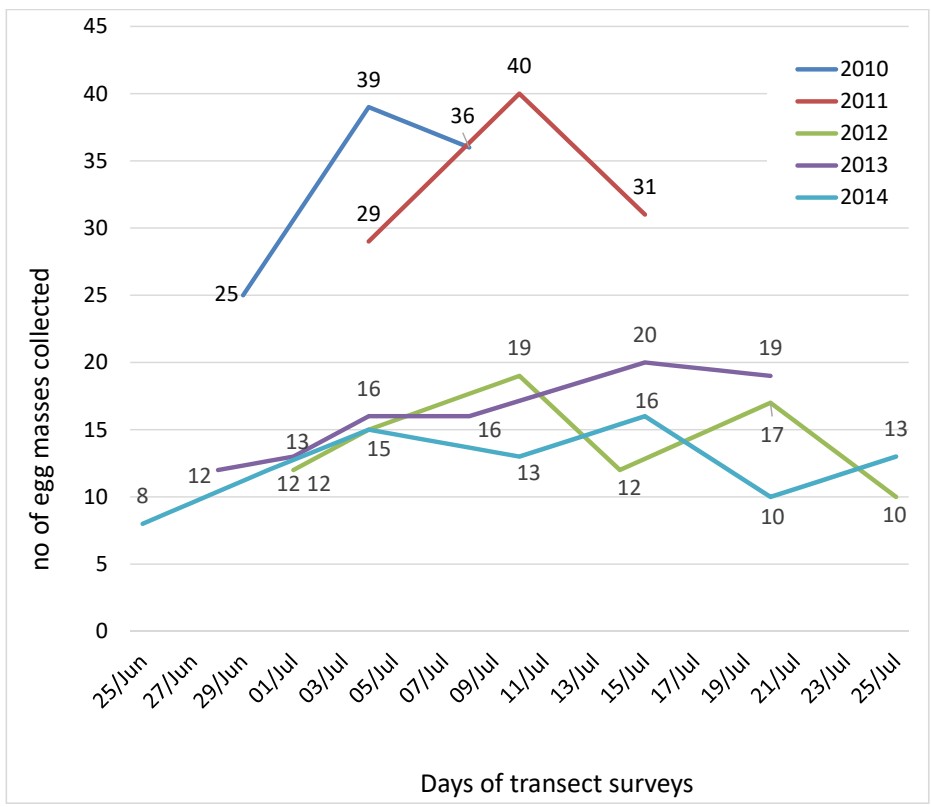

**Figure 2.** Number of *Thaumetopoea pityocampa* egg masses collected in each transect survey carried out on the cedar plantation of Chréa National Park, during the five years of the study.

The earliest start of the moth flight season was in 2014, when the first collection occurred on 25 June, and was also the year when more collection dates were needed (seven, the last made on 25 July). Six visits to the cedar plantation were made on 2012 (1–25 July) and in 2013, one week earlier (28 June–20 July).

The aim of collecting 100 egg masses was only achieved in the first two years (2010 and 2011) after three collections and the overall cumulated surveyed area was 7.5 ha. In the following years, more collections were needed and consequently twice the area needed to be surveyed (15 ha in 2012 and 2013; 17.5 ha in 2014). Considering the cumulated area surveyed in each year, the concentration of the egg masses was estimated (Table 1). A total of 468 PPM egg masses overall were obtained and 100,634 eggs were examined as well as egg mass length, twig diameter and egg parasitization rates.

**Table 1.** Descriptive analysis of *Thaumetopoea pityocampa* egg masses and diameter of their support cedar twigs (mean ± SD and min-max and mean parasitization percentage) and daily mean and maximum air temperatures (mean ± SD). Statistical significance of differences between years of study presented for each variable. Letters near the average represent LSD homogeneous groups after ANOVA analysis performed for each dependent variable related to years (grouping variable).

| Years | 2010 | 2011 | 2012 | 2013 | 2014 |
|---|---|---|---|---|---|
| Mean air temp. (°C) | | | | | |
| June | 19.4 ± 1.88 | 20.1 ± 1.68 | 23.7 ± 1.61 | 18.2 ± 1.96 | 20.1 ± 1.71 |
| July | 25.9 ± 1.43 | 23.8 ± 1.38 | 25.2 ± 1.39 | 23.3 ± 1.28 | 23.6 ± 1.44 |
| August | 24.2 ± 1.42 | 25.5 ± 1.34 | 26.9 ± 1.33 | 23.5 ± 1.18 | 24.8 ± 1.39 |
| Max. air temp. (°C) | | | | | |
| June | 26.3 ± 1.70 | 27.1 ± 1.53 | 30.2 ± 1.33 | 27.2 ± 1.77 | 27.6 ± 1.56 |
| July | 33.6 ± 1.52 | 32.1 ± 1.44 | 35.3 ± 1.39 | 31.6 ± 1.60 | 32.3 ± 1.90 |
| August | 31.9 ± 1.54 | 32.5 ± 1.61 | 36.1 ± 1.65 | 32.5 ± 1.44 | 33.5 ± 1.68 |
| Egg mass sample size | 100 | 100 | 85 | 96 | 87 |
| Egg mass density at breast height (per ha) | 13 | 13 | 6 | 6 | 5 |
| Egg mass length (mm) | 26.55 ± 4.83 [b] | 28.14 ± 5.92 [a] | 25.91 ± 4.80 [b] | 22.68 ± 3.76 [c] | 23.30 ± 3.96 [c] |
| | (18.0–38.0) | (16.2–46.3) | (17.0–41.0) | (17.0–39.0) | (17.9–41.1) |
| Twig diameter (mm) | 3.77 ± 1.04 [ab] | 3.95 ± 1.07 [a] | 3.68 ± 1.04 [bc] | 3.76 ± 1.05 [ab] | 3.27 ± 1.01 [c] |
| | (1.62–7.26) | (1.93–7.18) | (1.72–6.92) | (1.69–7.12) | (1.63–6.98) |
| Number moth eggs | 22,558 | 23,759 | 18,755 | 18,429 | 17,133 |
| Fecundity (moth eggs per mass) | 225 ± 47 [ab] | 237 ± 45 [a] | 220 ± 41 [b] | 191 ± 43 [c] | 196 ± 54 [c] |
| | (90–312) | (121–348) | (74–331) | (80–289) | (63–287) |
| Hatched eggs | 203 ± 61 (90.22%) [a] | 212 ± 64 (89.45%) [a] | 97 ± 93 (44.09%) [b] | 65 ± 64 (34.03%) [c] | 66 ± 60 (33.67%) [c] |
| | (27–312) | (15–334) | (0–276) | (0–229) | (4–242) |
| Unhatched egg | 12 ± 23 (5.33%) [a] | 17 ± 23 (7.17%) [ab] | 101 ± 95 (45.91%) [d] | 29 ± 34 (15.18%) [b] | 69 ± 42 (35.20%) [c] |
| | (0–160) | (0–130) | (0–273) | (0–244) | (2–177) |
| Parasitized eggs | 10 ± 27 (4.53%) [cd] | 9 ± 24 (3.86%) [d] | 23 ± 41 (10.33%) [c] | 98 ± 63 (51.14%) [a] | 62 ± 48 (31.58%) [b] |
| | (0–215) | (0–173) | (0–200) | (0–233) | (0–174) |
| Eggs parasitized by *B. servadeii* | 4 ± 9 (1.62%) [c] | 2 ± 8 (1.15%) [c] | 14 ± 37 (6.30%) [c] | 78 ± 69 (40.46%) [a] | 36 ± 42 (18.12%) [b] |
| | (0–49) | (0–54) | (0–200) | (0–233) | (0–126) |
| Eggs parasitized by *O. pityocampa* | 2 ± 11 (0.96%) [b] | 2 ± 8 (0.75%) [b] | 4 ± 20 (1.97%) [ab] | 6 ± 17 (3.07%) [ab] | 9 ± 25 (4.65%) [b] |
| | (0–89) | (0–173) | (0–179) | (0–96) | (0–119) |
| Eggs parasitized by *T. embryophagum* | 4 ± 24 (1.95%) [b] | 5 ± 22 (1.96%) [b] | 4 ± 11 (2.06%) [b] | 15 ± 38 (7.61%) [a] | 17 ± 42 (8.81%) [a] |
| | (0–215) | (0–54) | (0–80) | (0–196) | (0–174) |

From the egg masses, three parasitoid species emerged: Baryscapus servadeii (Domenichini, 1965) (Eulophidae), *Ooencyrtus pityocampa* (Mercet, 1921) (Encyrtidae) and *Trichogramma embryophagum* (Hartig, 1838) (Trichogrammatidae). After emergence, the exit holes of PPM larvae are large and circular, while Hymenoptera parasitoids found parasitizing PPM eggs are characterized by relatively smaller holes, especially those of *B. servadeii* and *O. pityocampa*. The third parasitoid found, *T. embryophagum*, has a very small exit hole, which varies in numbers, from one to three holes per egg (Figure 3). Unhatched eggs, recognizable by their whitish color, were opened to check for diapausing parasitoids or dead larvae.

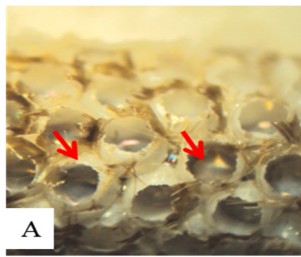 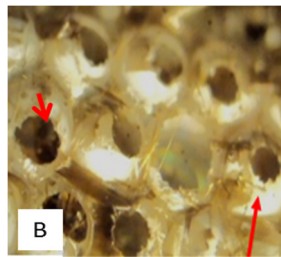 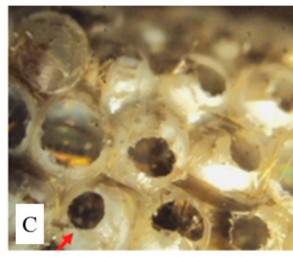 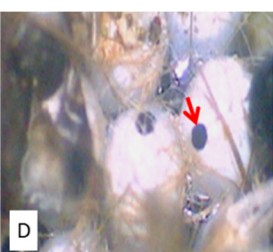

**Figure 3.** Exit holes of processionary moth larvae (*Thaumetopoea pityocampa*) (**A**) and Hymenoptera parasitoids species: *Baryscapus servadeii* (**B**), *Ooencyrtus pityocampa* (**C**) and *Trichogramma embryophagum* (**D**).

The number of eggs in the PPM masses varied significantly between years ($F_{(4,463)}$ = 107.4305; $p < 0.0001$), increasing from 2010 (maximum of 312 eggs/mass) to 2011 (348 eggs/mass), followed by a decreasing trend from 2012 to 2014, especially relevant in 2013 and 2014, where there was a maximum of less than 300 egg/mass. A similar trend was observed on the size of the egg masses, with statistically significant differences ($F_{(4,464)}$ = 21.8394; $p < 0.0001$) across years. Reproductive success observed in these two years was also revealed by a low number of unhatched eggs (averages of 5.33% and 7.17%, respectively), compared to the following years ($F_{(4,463)}$ = 53.8802; $p < 0.0001$). The significant decrease of the average number of eggs per mass and the number of hatched eggs after 2012 revealed the PPM population to decline; this can be mainly attributed to hatch failure observed in 2012, more than double that in previous years. There was also a decrease in the reproductive potential registered in 2013 and 2014. In the last year of this study, not only were there the lowest egg mass densities but the egg masses were also placed by PPM females on the thinnest twigs, being the main variables responsible for the statistically significant differences observed between years ($F_{(4,464)}$ = 5.0090; $p = 0.0002$).

The average number of eggs per mass remained high (191 to 220), and the percent of eggs that were unhatched was also high, mainly in 2012 (45.91%) and 2014 (35.20%). In 2013, the unhatched rate remained high (15.18%), twice that observed in 2010 and 2011, but this year was also distinguished by the highest parasitic rate, reaching an average of 51.1% of the eggs, statistically different from other years ($F_{(4,463)}$ = 73.0526; $p < 0.0001$) (Table 1). In fact, 2012 was by far the hottest year in the months when the adult moth was active and producing the egg masses, with mean air temperatures ranging from 23.7 °C in June (more 3.6 °C than in the other years of the study) to 26.9 °C in August, (more than 1.4 °C) and mean maximum temperatures over 35 °C in July and 36 °C in August. These temperatures may have contributed to the high rate in failure of embryonic development observed in 2012, with negative impacts in the PPM population in the following years, expressed in the decrease of egg masses found. 2013 was the coolest year, which may have contributed to the high parasitism rates registered. In fact, moth egg production per mass decline in 2013 and 2014 may have resulted from the combination of many eggs failing to hatch (35% unhatched) and high parasitization (31% of all eggs were parasitized, with the highest levels observed for *O. pityocampa* and *T. embryophagum*).

*B. servadeii* revealed to be the most important parasitoid species, showing major contributions as the PPM population declined ($F_{(4,463)}$ = 59.4432; $p < 0.0001$).

During 2010 and 2011, when it was possible to collect the targeted 100 egg masses and the fecundity rates were the highest, the parasitization rate of eggs were less than 2% for

each of the three parasitoid species (progressive phase of parasitoid populations). However, the parasitoid populations significantly increased the following years, with *B. servadeii* assuming a major role, in relation to the other two species. An overall average per species of the parasitisized eggs of 6.76% was obtained for the five years of the study, and this value was only surpassed in the last two years (2013 and 2014), and only by *T. embryophagum* (slightly) and *B. servadeii*, reaching an impressive 40% of the PPM eggs (representing almost 80% of the parasitized eggs) in 2013, surely contributing to the reduction of the PPM population in the following year (only 87 egg masses collected). The impact of parasitoid species varied among the years, even for the less abundant species: *T. embryophagum* ($F_{(4,463)}$ = 4.1149; *p* = 0.0028) and *O. pityocampa* ($F_{(4,463)}$ = 2.7508; *p* = 0.0277), revealing a complex dynamic interaction between the availability of moth eggs and the different parasitoid species' responses.

The effect of the parasitoids gradually intensified during the moth population regression phase, with parasitization rates increasing mostly as a result of *B. servadeii*.

The accumulated results of the egg masses and the emergence of parasitoids over the years of the study revealed the temporal synchrony between PPM and its egg parasitoids as expected. Emergence dates of the three parasitoids occurred mainly in July. *T. embryophagum* and *O. pityocampae* were observed earlier (first specimen emerged 7 to 8 days after egg mass collection), starting in the second week of July, and *B. servadeii* occurred a few days later (first specimen obtained 9 days after egg collection) and matched the PPM larvae increase. In fact, the majority of the PPM eggs that were not parasitized hatched later, during August.

## 4. Discussion

The temporal variation of PPM populations in the cedar forest can be characterized by transcyclical fluctuations imposed by biotic and abiotic factors [26–28]. The average number of eggs laid per female also expresses the energy allocated to reproduction [29,30] and reflects, consequently, the potential of the pest. However, an increase in egg production will eventually reduce the available resources [31–33], since the optimization of reproductive effort, which tends to maximize survival, depends on the quantity and quality of nutrient reserves [34,35], and may have contributed to the decline of the moth population observed after 2012.

In the Atlas Blideen cedar forests, the population reached its peak between 2011 and 2012 when it was possible to collect the targeted 100 egg masses and maximum numbers of eggs per egg mass. The laying size, defined by the number of eggs laid per act [26,36], is linked to the female's egg stock and their physiological condition [26]. These intrinsic parameters in the presence of favorable conditions, in particular, resource availability, allow for a high reproductive effort and consequently increase the dispersal and colonization capacities. At this stage, the effect of natural enemies and especially parasitoids, is limited despite their temporal and spatial synchronism.

The main factors responsible for the failure of embryonic development observed in the later years of the study could be the high temperatures during summer of 2012 and resources becoming a limiting factor due to high density of moth larvae, affecting both the quality and the quantity of egg masses [37]. Under these conditions, temporal and spatial availability of resources becomes a survival factor, which can induce changes in the capacities of the insect, namely in the amount and fecundity of egg masses [38,39].

Climate change is causing changes in the distributions of species around the world, with an overall shift away from the equator and towards the poles, invading new areas and causing the displacement of natural enemies. Currently available climate models predict that the Mediterranean area is one of the hot spots of climate change [40]. Studies carried out in Portugal and Tunisia proved that temperatures in the range registered in July and August 2012 (maximum temperature above 36 °C) cause high embryo mortality in the PPM [41], contributing to the high percentage of unhatched eggs observed. On the other hand, in the epidemic phase, competition arising from high densities has delayed effects on adult performance [31]. These unfavorable interactions directly influence the bio–demographic

characters and affect the survival, the growth and reproduction of individuals [42] and consequently leads to a decrease of the population [43–46], observed in our study on 2013.

This weakening of the moth populations is accentuated by the action of natural antagonists where the abundance and diversity of egg parasitoids play an important role [47], which was more noticeable on the regression phase, after 2012, and it reached its maximum in 2013 with an average of 100 parasitized eggs per clutch. The activity of parasitoids, mainly *B. servadeii* and *T. embryophagum*, continued in 2014, but with less impact. Our results are in agreement with Battisti [48] and Mirchev et al. [49] which concluded that the impact of parasitoids is more relevant before and after the peak in PPM populations.

Of the parasitoid species found in our study, *B. servadeii* and *O. pityocampae*, are the most abundant and best studied. *B. servadeii* is a specialist of PPM while *O. pityocampae* is a generalist egg parasitoid, requiring an alternant host species to complete its first generation before PPM eggs are available [50]. *T. embryophagum* has a wide host range and is frequently mass reared for augmentative biological control of many insect pests in agricultural and forestry environments [51,52].

*B. servadeii* is abundant during all the host population phases because its parasitic action on PPM eggs facilitates its infiltration through the protective scales [53,54]. This effect was observed by Geri [14] in Corsica Mountains, Bellin et al. [55] in Greece and Schmidt et al. [3] in Morocco. Tiberi [56] reported strong activity of this parasitoid only in the moth decline phase. Zamoum et al. [57] linked the dominance of *B. servadeii* to the abundance of egg-laying and the structure of the pine forest stand.

On the other hand, the polyphagous behavior of *O. pityocampae* promotes the dispersion of its action presenting a more active parasitism in the decline phase in Chréa. However, in Italy, Tiberi [56] noted its predominance in the moth increase phase, while in the Aleppo pine at Djelfa, Zamoum et al. [57] described a regulatory effect of *O. pityocampae* in relation to the abundance laying of processionary moth. Mirchev et al. [58] noted its dominance compared with *B. servadeii* in Turkey. As for the activity of *T. embryophagum*, it is more important after the host population's peak where its regulatory effect is significant, although the number of parasitized egg masses remains low. Petrucco-Toffolo et al. [59] attribute this constraint to the protective scales that hinder the parasitic success. However, the results of Daniel et al. [60] have reported an abundance of this species in Italy.

Our results highlight a spatiotemporal structuring between *Trichogramma* and *Ooencyrtus* where the abundance fluctuations of these species are inversely correlated. This relationship is a strategy to avoid super parasitism by species with the ability to distinguish parasitized hosts [61]. In fact, Battisti [48] note that *O. pityocampae* is more active on older eggs, while *B. servadeii* prefers freshly deposited eggs, which may explain the higher parasitism rates observed in this study in June than in August.

## 5. Conclusions

Temporal variability of the processionary moth populations results from its adaptive capacities to environmental changes. Natural regulation of the PPM on Atlas cedar is ensured by the simultaneous presence of multiple entomophagous species, namely three species of egg parasitoids: *B. servadeii*, *T. embryophagum* and *O. pityocampae*. These parasitoids are characterized by fluctuating efficiency depending on the spatiotemporal distribution of the PPM host populations. In Chréa area, *B. servadeii* is the most abundant and impactful on PPM egg survival, while parasitization by *T. embryophagum* and *O. pityocampae* provide additional contributions to egg mortality, despite the obstacles caused by the coverage of scales. Temperature is also an important extrinsic abiotic factor and the high rate of unhatched eggs registered in 2012 may be an effect of the hottest summer registered during the five-year study.

**Author Contributions:** Conceptualization, investigation and writing, S.S.; review, editing and statistical analysis, L.B.; supervision and reviewing, G.C. All authors have read and agreed to the published version of the manuscript.

**Funding:** Publication was supported by GREEN-IT Bioresources for Sustainability/ FCT, IP (Portugal).

**Data Availability Statement:** The data that support the findings of this study are available from the corresponding author upon reasonable request.

**Acknowledgments:** The authors thank Richard Hofstetter from North Arizona University for final revision of the manuscript.

**Conflicts of Interest:** The authors declare no conflict of interest.

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
