# Peer review of "Role of Egg Parasitoids in Controlling the Pine Processionary Moth in the Cedar Forests of Chréa National Park (Algeria)"

_forests, doi:10.3390/f13020211_

Round 1

Reviewer 1 Report

This is an interesting work showing the population dynamics of the pine processionary moth and their egg parasitoids through several years in a National Park. However, I am not sure that the results are reflecting the aims (and the title) of the study: “The main objective of this study is to derive the most useful information on the natural process of PPM population regulation, and to understand the functional response of oophagous parasitoids, their efficacy and their ecological role in reducing the impact of the moth on cedar stands with the National Park”. Regarding the synchrony in the maximum of parasitism and the minimum in the egg production, there is not enough explanation for the population dynamics of both pest and parasitoids. Furthermore, half of the introduction section is dedicated to climate change and how it will affect the PPM but no climatic data are provided for the study, which I think will help to explain some of the results. My recommendation is that authors should limit themselves to the results obtained, and to include climatic variables in the analyses, avoiding speculation. Here are some other more specific comments thay may help:

Line 68: maybe you mean“within the National Park”?

Line 79: coordinates should be located when you mention the Park, I suggest lines 71 or 72.

Please indicate the type of trap used, how many were set and were (e.g., in the cedar branches, at 1.5 m high). Also indicate how many transects were surveyed.

Figure 2 and lines 102-109 are results, since you did not know a priori which parasitoid species you were going to find. Please place it in that section.

In data analysis you need to indicate clearly what are you going to compare with the ANOVA. And I do not see you are statistical analyzing “the effect of oophagous parasitoids”. Please revise this section.

Figures 4 and 5 are redundant with table 1, if you want to include these figures you should not repeat that information on table. In that case, please be aware that figure 5 legend does not correspond with what you are showing in the figure: in the legend says “percentage” but in the graph says “mean number”.

In lines 173-180, you relate the unhatched eggs proportion with temperature, but no climate data are provided. You should include them in order to support these statements. Also, in the last sentence of that paragraph you say that “moth egg production per mass decline in 2014 was a combination of many eggs failing to hatch and many were parasitized”. This is incorrect: those are causes for a low moth emergence, but egg production is prior to those events, and they will very likely influence the next year’s egg production. Furthermore, the synchrony in low egg production and the high parasitism is very anomalous, there should be a delay in the low moth population from the high parasitism. Therefore, the peak in parasitism, especially in T. embryophagus can not be explained just by the PPM populations, even more when this species is polyphagous.

I do not understand the last paragraph of your results. You are saying “The emergence dates of the three parasitoids occurs mainly in July, when most of the egg are parasitized”. That is pretty obvious. And these results do not represent a temporal synchrony since you are evaluating parasitoids emerging from those eggs!! You have no enough data for that statement. And in the same paragraph “egg” is written in several occasions instead of “eggs”.

Lines 247-248: “According to our results global warming may have a negative effect, as approaches the high temperature threshold of embryonic development”. You have not provided temperature data and this is merely speculative.

Author Response

Dear reviewer,

Thank you very much for the time spent in reviewing the manuscript and the helpful comments that improved it considerably. Hope the questions and doubts are clarified.

Reviewer 2 Report

 Reviewer 1X; Forests 2021, 12, x.

 The manuscript ¨ The role of embryonic parasitoids in the preservation of Cedar forests against cyclical outbreaks of the pine processionary moth in Chréa National Park (Algeria)¨, has information of some natural control factors of a lepidopteran pest on cedar in a park in Algeria. The information could be relevant for the northern Mediterranean region in places that share similar environmental conditions. Although the manuscript has some value, authors need to make several important changes before it has the quality for publication in this journal. In summary, some details that must be addressed to improve the manuscript are: 

·       The overall quality of the English is bad, I did not spend time trying to improve sentences that are redundant, use a lot of verbosity or need improvement in scientific style. Also, there are a few grammar mistakes and sentence structure that should be improved. I marked some of this comments in yellow in the manuscript (pdf file) and comments below, but there is a lot of work to do, I just pay attention to these deficiencies in the first two sections of the manuscript (Introduction, Materials and Methods), but all the manuscript must have the review of an expert (it is not the job of the reviewer). I think that the MS would benefit from being proofread by a native English speaker who knows the scientific writing style.  

·       The objectives as they were wrote were not met. Authors exaggerated things with the data they presented, and the inference they achieved. It is possible that there is a problem with the language to express what they did. The objective must be rewritten. 

·       This manuscript was not developed to demonstrate any Climate Change (global warming) effect on entomology. There is an international consensus on this global problem, but authors should decrease the number of times they mention it, and they should concentrate on explaining the results with data that they can only contrast (avoid speculation).

 ·       The authors need to clarify some methodological details. Some lines of discussion were mixed in the result section. The conclusion section must be rewritten. There is too much speculation in there.

Complementary comments 

Title. Title should be change. The number of words should be reduced, the most common names used in the biological control area should be included.  Suggestion: Role of egg parasitoids on the Pine Processionary Moth population in Chréa National Park (Algeria)

 L14. …"through ecological stability of the ecosystem."; "ecological" seems redundant in the sentence. Suggestion: through stability of the ecosystem.

L16. … "Embryonic parasitoids" should be written "Egg parasitoids". Egg parasitoids is the most often name for these parasitoids used in classic biological control books and papers.  

L20…. "of three". It would be better if replaced by …of the…

L28-29. How should scientific names be written? Italics?

L 40. … "Its cyclical epidemics affect the health status of forests trees," Correct sentence (grammar and style): It affects the health status of forest trees,

L43. … "with large outbreaks"  Correct sentence (style): with often outbreaks 

L 45-47. … The sentence should be rewritten (It seems redundant to me).

L55. … "to higher elevations and latitude". Correct sentence (style): to higher altitude and latitude.

L62-65. … "The development of control programs must integrate the ecosystem as a whole into the overall ecological control strategy in order to establish sustainable management of infestations in protected sites. " Correct the sentence (It is a sentence with too much verbosity).  

L65-68. …"The main objective of this study is to derive the most useful information on the natural process of PPM population regulation, and to understand the functional response of oophagous parasitoids, their efficacy and their ecological role in reducing the impact of the moth on cedar stands with the National Park. " Correct the sentence (it is a sentence with too much verbosity). Also, authors must write only what they could contrast with data. Authors did not show anything of funtional response, and the ecological role of natural enemies was limitated as they did not evaluate other mortality factors of the pest (larvae, pupae, adults).  The ecological role of egg parasitoids may be overestimated, this egg mortality may be a type of replaceable mortality (there are other mortality factors in larvae of the first instars that could reduce the relevance of the egg parasitism). It seems that this work only determined the percentage of parasitism of eggs of the PPM and related it to the temperature in five cycles (years). That is what should be established in the objective. 

L76-77. "is 5°C in winter. " A measure of dispersion in temperatures must be included (standard deviation).

L81. "Cedar plantation".

L84. "at breast height" Correct the sentence; t is better to set the height in meters.

L85-88. How much time was spent collecting egg masses in each year?

L94. "Tubes are numbered and placed" Correct the sentence (grammar): Tubes were numbered and placed

L95. "ambient conditions" ¿? Do you mean room temperature?

L96. "which taken to the" Correct the sentence (grammar): which were taken to…

L98-99. "protective scales (that were detach from the abdomen of the female to cover the eggs)" Correct the sentence: do not used unnecessary expressions: protective scales of the eggs were removed…

L102. processionary moth Correct: PPM or T. pityocampa.  

L116. "to a series of statistical analyzes" Correct, what do you mean?

L117. "oophagous parasitoids" Correct: egg parasitoids. 

From now on, I will no longer comment on bad writing (although I marked yellow in the manuscript), it is the responsibility of the authors and the editor to ensure that this is corrected or the manuscript should be rejected.  

L147. Number of eggs per mass. When authors include a measure of central tendency (media) they should include a measure of dispersion (standard deviation) in the whole manuscript.

L173-179; L204-207; L208-2015. Be careful, you are mixing up some discussion in the results section. 

Figure 6. It is not very useful. Remove it. 

L226. There seems that there is some speculation in here.

L247-248. There seems that there is some speculation in here.

L263. What do you mean?

L293. There are too much speculation in the Conclusions section. Authors must conclude only in the base of date they presented. 

Author Response

Dear reviewer

Thank you very much for the time spent in reviewing the manuscript and the helpful comments that improved it considerably. Hope the questions and doubts are clarified.

Round 2

Reviewer 1 Report

This version as significantly improved and it fits my expectations. I only have minor comments:

Line 22: it is not distribution, I guess you mean abundance, or population, or proportion.

Line 144: “… were placed in pairs…”

Line 149: “… more abundant, and therefore only three surveys were needed…”

Line 348: do not include the number of parasitoid species, in materials and methods you do not know yet how many species you are going to find.

Glad to see you have included weather data in your results, but you also need to indicate it in materials and methods section, and to state where did you get the data from.

Line 442: I suggest to start the paragraph by saying the parasitoids you foung, for instance: From the egg masses, three parasitoid species emerged: Baryscapus servadeii (Domenichini, 1965) (Eulophidae)…

Lines 447-448: please be sure that those data about what happened to the unhatched eggs is included somewhere.

Lines 486-491: you are repeating some data from the previous paragraph, avoid redundance.

Line 539: “… (more than 1.4ºC)…”

Line 548: “… the most important parasitoid species…”

Line 565: please delete the comma before the parentheses.

Line 619: “…will eventually reduce in the available…”

Line 628: I am not a native English speaker, but I think it should be “… allows for a high reproductive effort…”. Please revise.

Author Response

Thank you very much for the time spent in reviewing again the manuscript and we are very pleased that we have accomplished your expectations. 

Attached you'll find our reply to the new comments. 

Reviewer 2 Report

The manuscript improved. 

I am agree that all the comments have been addressed in the manuscript:
Role of egg parasitoids in controlling the pine processionary moth in the cedar forests of Chréa National Park (Algeria)

I found a couple of minor mistakes. Please, take care of them.

Page 7,  paragraph before discussion:

Line 605...and its embryonic parasitoids as expected.

I should be written: ...and its egg parasitoids as expected.

Page 8, second line of the third paragraph:

a space between words is missing.

Page 9, four line of the first paragraph:

I think there is and unneccesary ¨s¨ in the sentence.

Author Response

Thank you very much for the time spent in reviewing again the manuscript and we are very pleased that we have replied in accordance to your comments. 

Attached you may find the reply to the new comments. 
